# STRAS: a new high time resolution aerosol sampler for PIXE analysis

Silvia Nava[1,2], Roberta Vecchi[3,4], Paolo Prati[5,6], Vera Bernardoni[3,4], Laura Cadeo[3], Giulia Calzolai[2*], Luca Carraresi[1,2], Carlo Cialdai[2], Massimo Chiari[2], Federica Crova[3,4], Alice Forello[3,4], Cosimo Fratticioli[1,2], Fabio Giardi[2], Marco Manetti[2], Dario Massabò[5,6], Federico Mazzei[5,6], Luca Repetto[5], Gianluigi Valli[3,4], Virginia Vernocchi[6], Franco Lucarelli[1,2]

[1] Dept. of Physics and Astronomy, University of Florence, Sesto Fiorentino, I-50019, Italy;
[2] National Institute for Nuclear Physics, INFN-Florence, Sesto Fiorentino, I-50019, Italy;
[3] Dept. of Physics, University of Milan, Milan, I-20133, Italy;
[4] National Institute for Nuclear Physics, INFN-Milan, Milan, I-20133, Italy;
[5] Dept. of Physics, University of Genoa, Genoa, I-16146, Italy;
[6] National Institute for Nuclear Physics, INFN-Genoa, Genoa, I-16146, Italy;
*Correspondence to*: Giulia Calzolai (calzolai@fi.infn.it)

**Abstract.** The joint use of hourly resolution sampling and analyses with accelerated ion beams such as *Particle Induced X-ray Emission* (PIXE) technique has allowed the measurement of hourly temporal patterns of particulate matter (PM) composition at many sites in different parts of the world. The demand within the scientific community for this type of analysis has been continuously increasing in recent years, but hourly resolution samplers suitable for PIXE analysis are now discontinued and/or suffer from some technical limitations. In this framework, a new hourly sampler, STRAS (*Size and Time Resolved Aerosol Sampler*), was developed for the collection of $PM_{10}$, $PM_{2.5}$ or $PM_1$. It allows automatic sequential sampling of up to 168 hourly samples (1 week), it is mechanically robust, compact, and easily transportable. To increase PIXE sensitivity, each sample is concentrated on a small surface area on a polycarbonate membrane. The comparison between the elemental concentrations retrieved by STRAS samples and samples collected using a standard sequential sampler operated in parallel shows a very good agreement; indeed, if both the samplers use the same kind of membrane, the concentrations of all detected elements are in agreement within 10%.

## 1 Introduction

Atmospheric particulate matter (PM) has been shown to represent a significant concern due to its negative effects on human health, air quality, and visibility around the world (WHO, 2021; IPCC, 2023). The complexity of atmospheric aerosol properties and the variety of emission sources claim for advanced experimental and modeling techniques for a deeper understanding and more robust data interpretation and source apportionment.

Many studies have been devoted to PM sampling, analysis and source apportionment using 24-h averaged data, in order to have sufficient quantities of material to perform a chemical characterization as complete as possible, and thus to obtain indications on the predominant sources of PM. However, most particulate emissions as well as atmospheric dilution processes change within a few hours. Thus, daily sampling cannot track these rapid changes. As some source emissions can heavily

affect air quality with very high loading of toxic elements during a few hours, the knowledge of the timing and the intensity
of specific episodes may be important for the assessment of human exposure as well as for source identification and
apportionment. Furthermore, source apportionment receptor models (Belis et al., 2019) require a series of samples containing
PM material from the same set of sources in different proportions: increasing the measurement time resolution typically results
in samples with greater between-sample variability in the source contributions compared to samples integrated over longer
time periods (Crespi et al., 2016; Crova et al., 2024).
High-time resolution resolved measurements (i.e. less than a few hours) require suitable techniques both for particle sampling
and analysis. For time-resolved sampling one of the most suitable solutions is to use a system that automatically switches the
particle collection substrate, for example every hour, to obtain a sequence of high-time resolution collected aerosol
spots/deposits. However, this leads to a large number of "small" (as for mass and deposit area) collected samples to be
analyzed. In this frame, Particle Induced X-ray Emission (PIXE) technique is an optimal analytical solution: using a properly
collimated proton beam, a high number of aerosol deposits can be quickly analyzed, without any sample preparation. This
technique is non-destructive, multi-elemental (it allows the simultaneous detection of all the elements with $Z>10$) and very
fast, with a high sensitivity also for small amount of matter. The signal, the X-ray yield, is indeed proportional to the areal
density of the deposit: even a few micrograms can be sufficient if concentrated on a small ($< 1$ cm$^2$) area (Calzolai et al., 2015).
Other widespread analytical techniques like Ion Chromatography and Inductively Coupled Plasma Mass Spectrometry cannot
be used to analyze this kind of samples: the only possible alternative is the use of advanced energy dispersive X-ray
fluorescence (EDXRF) spectroscopy with collimated/micro irradiating beams or synchrotron radiation XRF, but beam
availability has largely limited the development and use of this technique. It is noteworthy that among PIXE detectable
elements there are markers of specific components such as marine aerosol (Na, Cl), mineral dust (Al, Si, Ca, Ti, Sr), sulphates
(S), biomass burning (K), heavy oil combustion (V, Ni), traffic and industrial emission (Mn, Ni, Cu, Zn, Pb), etc.
At the INFN LABEC ion beam laboratory (Chiari et al., 2021), a dedicated PIXE set-up using a proton beam extracted into
ambient pressure (external beam) has been optimized for the analysis of aerosol samples. Moreover, LABEC hosts the
Elemental Mass Calibration Centre, EMC2, the reference center for measuring the mass concentration of particulate elements
within the pan-European Aerosol, Clouds and Trace Gases Research Infrastructure, ACTRIS (https://www.actris-ecac.eu/).
The advantages of such a set-up are the very short measuring time (30/60 seconds/sample) and the possibility to analyze
samples with very low mass such as those obtained with high-time resolution or size segregated samples (Lucarelli et al., 2014;
Calzolai et al., 2015; Lucarelli et al., 2018).
Traditionally, the high time resolution sampling has been carried out using streaker samplers by PIXE Int. Corp. (see e.g.,
D'Alessandro et al., 2003). Briefly, in this device, atmospheric particles are separated on different stages by a pre-impactor
and an impactor, at a flow rate of 1 l min$^{-1}$. The pre-impactor removes particles with aerodynamic diameter $D_{ae} >10$ μm. The
aerosol coarse fraction (2.5 μm $< D_{ae}< 10$ μm) deposits on a Kapton foil which acts as the impactor stage, while the fine
fraction ($D_{ae} < 2.5$ μm) is collected on a polycarbonate membrane. The two collecting substrata (Kapton and polycarbonate)
are paired on a cartridge which rotates at a constant speed for one week: this produces a circular continuous deposition of

particulate matter ("streak") on both stages. PIXE analysis of these samples by a properly collimated proton beam, which scans the deposit in steps corresponding to 1-h of aerosol sampling, provides the elemental concentrations with hourly time resolution (Lucarelli et al., 2014; Calzolai et al., 2015; Lucarelli et al., 2018). The "streaker+PIXE" approach has demonstrated its effectiveness in many studies (see for example: Prati et al., 2000; D'Alessandro et al., 2003; Crespo et al., 2010; Amato et al., 2011; Dall'Osto et al., 2013; Nava et al., 2015; Lucarelli et al., 2019; Forello et al., 2019), however this sampler, designed about 40 year ago (Annegarn et al., 1988), is now discontinued.

On-line high-time resolution instrumentation for the comprehensive analysis of aerosol composition (non-refractory components, ions, carbonaceous components, elements) is nowadays more and more used in monitoring campaigns and available for detailed source apportionment studies. As for the elemental concentrations, which are the focus of this work, a near-real-time multi-elemental ($Z>12$) monitor, the Xact® 625i (by Sailbri Cooper Incorporated - SCI), has been developed. This device uses reel-to-reel filter tape sampling and energy dispersive XRF analysis: the PM is sampled using a low volume PM size-selective inlet (working at 1.0 $m^3h^{-1}$), typically for 1-4 hours, and the resulting deposit is then advanced into the measurement chamber where it is analyzed by EDXRF for selected elements while the next sample is collected. Xact® 625i has shown good results and its use is expanding in the aerosol research community. Nevertheless, Xact® 625i showed over/underestimation of the concentration of different elements compared to reference methods (Furger et al., 2017, Tremper et al., 2018; Zhu, Y. et al., 2024), the detection of light elements like Na and Mg (typical markers of fresh and aged sea salt) is not possible and also the sensitivity for Al and Si (typical markers of soil dust) is lower compared to PIXE. In addition, the instrument price limits the possibility of campaigns with multiple sites in parallel, which can instead be achieved through the use of multiple cheaper samplers and subsequent PIXE analysis in one central laboratory. Finally, the X-ray source usage on the field is often subject to radioprotection restrictions by law.

In this framework, the new high-time resolution sampler STRAS (*Size and Time Resolved Aerosol Sampler*) has been developed, as described in this paper, to replace the streaker sampler while improving the performance and technical characteristics of that previous sampler.

In extreme summary, STRAS allows the following improvements with respect to the (discontinued) streaker: higher areal load (and so higher sensitivity for PIXE/EDXRF/optical techniques); flexibility in time-resolution; possibility to work directly with commercial EPA inlets; data logging for sampling parameters and for sample position.

Further, it has to be noted that STRAS allows the collection of 1 week of hourly samples without any operator intervention (or filter change); as a comparison, most of the automatic aerosol samplers allow to load up to about 20 samples.

A thorough discussion of these issues is given in the following sections.

This device has been designed with the aim of obtaining a robust, reliable high time resolution sampler for subsequent PIXE analysis. In fact, with respect to most of the analytical techniques (such as ICP, IC, and conventional EDXRF), PIXE is much faster (allowing measurements with very good DLs with a measurement time down to 30 sec, with no need of sample pre-treatment) and thus it is the main technique applicable to such a large number of samples. As aforementioned, EDXRF with collimated/micro irradiating beams or synchrotron radiation could be an option, but not easily available. Further comparison

among the analytical techniques goes beyond the scopes of this paper but may be found in literature (e.g., Calzolai et al., 2008;
Traversi et al., 2014; Yatkin et al., 2016).

## 2 Sampler design

The sampler has been conceived and designed in order to obtain a high-time resolution (modulable from 30 min to a few hours)
PM10, PM2.5, and PM1 collection, on "small" (about 1 cm$^2$) deposit areas, with automatic sequential sampling of at least 168
samples (1 week of hourly samples), to be mechanically robust, easy to use, compact and easily transportable.
The deposit area of each hourly sample should be small for two reasons. The first is that, as already mentioned, in PIXE spectra
the signal (counts in the area of the fluorescence X-ray peaks) is proportional to the areal density of the deposit, while the
continuum background is mainly due to secondary-electron bremsstrahlung radiation emitted by the collecting substratum:
increasing the deposit-to-substratum thickness ratio results in a higher signal-to-noise ratio. The second reason is the necessity
to limit the size of the sampler using a compact geometry: if 168 deposits are placed on a single sampling substrate compactly
(like in the streaker sampler), this reduces the size of the sampler and also facilitates both sampling automation and subsequent
PIXE analysis.
One of the best sampling membranes for PIXE analysis is the polycarbonate filter: it is made of very light elements which are
not detectable by PIXE, it generally presents very low background values, and it is thin (about 10 μm), thus producing a low
secondary electron bremsstrahlung background in PIXE spectra (Calzolai et al, 2015). It is indeed the filtering material so far
used in the streaker sampler.
The deposit thickness in terms of PM areal density (μg cm$^{-2}$) is given by the product of the PM concentration in air (μg m$^{-3}$),
the sampling time and the face velocity, which is equal to the flow rate divided by the deposit area. To use the sampler with
easily available commercial inlets, one possible choice is a flow rate of 2.3 m$^3$ h$^{-1}$ (European standard) or 1 m$^3$ h$^{-1}$ (US
standard); due to the necessity of a small deposit area, we opted for the solution with the lower flux (1 m$^3$ h$^{-1}$, i.e. 16.7 lmin$^{-1}$)
to limit the pressure drop across the filter and the risk of filter clogging. At this flow rate, using a deposit area of 0.90 cm$^2$, the
face velocity is about 309 cm s$^{-1}$, which is about three times the one used in the filtration stage of the streaker sampler (1 l min$^{-1}$,
$^1$, 2x8 mm$^2$ `nozzle: v ≈ 104 cm s`$^{-1}$): in these conditions, the areal density of the deposit is tripled and we obtain a factor
three gain (G=3) with respect to the streaker.
However, with these flow rate and deposit area, the pressure drop on a polycarbonate membrane can easily reach relatively
high values (>50-60 kPa), which can become difficult to sustain in continuous sampling campaigns. This difficulty has been
addressed by following two approaches. First, the possibility of increasing the deposit area to 1.2 or 1.5 cm$^2$ was implemented
in the STRAS, with the gain factor changing to G=2.2 or 1.8, accordingly (the same effect can also be reached keeping the
0.90 cm$^2$ area and reducing the flow rate to 0.75 and 0.60 m$^3$ h$^{-1}$, respectively, provided that the inlets are adapted accordingly).
Second, 0.8 μm pore size polycarbonate® membranes (Sterlitech) were selected instead of the more used 0.4 μm pore size
ones (the problem of possible PM losses due to this choice is discussed in section 4.1).
As a consequence of these experimental observations, the pressure drop was taken into account in the choice of the pump to
be used for sampling; in the final design, the dry vacuum pump (Becker VT 4.8) was considered the best choice in terms of
performance and manageability.
A cylindrical geometry, similar to that of the streaker, was chosen to minimize overall dimensions. A circular filter with a
radius of 14 cm makes it possible to collect 168 samples, each one with dimensions of 3 cm (along the radial direction) by 3
to 5 mm, corresponding to a deposit area from 0.9 to 1.5 cm$^2$. The filter is housed in a sealed cylindrical aluminum chamber
(Fig. 1): the air enters from an inlet in the chamber cup, passes through the filter and exits from the suction nose, which is
placed immediately under the filter, in contact with it to avoid flow losses (as can be verified by observing the pressure drop
on the filter itself). The dimensions of the suction nose define the shape of the deposit.
The filter is coupled to a rotation system, which allows its automatic movement at every programmed time interval (1 hour, or
other time intervals) using a controlled stepper motor (actually, the same rotation system used at the external beam PIXE set-
up at LABEC). Sampling intervals longer than one hour may be useful in remote areas, both to collect enough PM and to
increase the sampler's autonomy (for example one month with 4-h resolution).
As regards the rest of the sampling system (inlet, cabinet, pump, air flow control system), thanks to the collaboration with the
Italian Company Dadolab s.r.l. (Cinisello Balsamo, Milan, Italy), it was decided to integrate STRAS in their sequential sampler
Dadolab Giano$^{TM}$ (https://www.dadolab.com/en/products/ambient-samplers/pmx-giano-gemini), replacing its sequential
sampling system with the STRAS chamber. Particular attention was paid to integrate data logging for sampling parameters
(air flow, pressure drop, elapsed time etc..) and for sample position (the position of every single spot is registered and can be
reproduced also in case of black-outs or stepper motor malfunctioning). While unattended, STRAS can be controlled from
remote. STRAS was designed to make it possible the use of EPA inlets ; we used commercial single cut-off impactors operating
at 1m3h-1 with flow regulated within 2% (available for PM10, PM2.5 and PM1).
The sampler can be easily moved, the total weight is about 36 Kg and the dimensions of the cabinet are 70 cm x 50 cm x 30
cm.

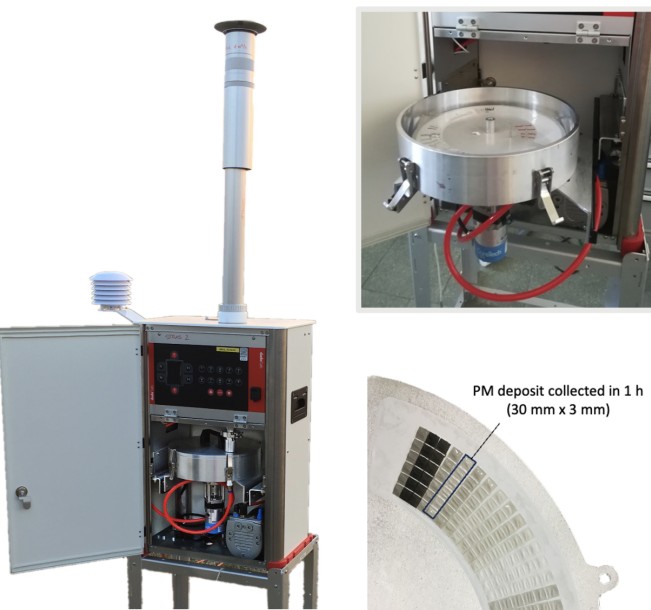


**Figure 1: The STRAS sampler. On the left, a picture of the whole sampler: the chamber with the rotating system can be seen inside the cabinet. On the right, a picture of the chamber when it is extracted (using linear guides) to be opened for inspecting or changing the filter. Below, a picture of a portion of the collecting polycarbonate membrane, where aerosol deposits can be seen.**

## 3 PIXE analysis of STRAS samples

Samples collected by STRAS can be easily analyzed by Ion Beam Analysis (IBA) techniques, like PIXE. In Fig. 2, the analysis of a STRAS sample at the INFN LABEC external beam PIXE setup is shown. The ion beam, typically a proton beam, can be collimated to match the aerosol spot collected in 1-h, and possibly scan the spot area. No sample pre-treatment is needed. In our setup, the proton beam is extracted in air through a 500 nm $Si_3N_4$ window and the samples are positioned at about 1 cm distance from it, perpendicularly to the beam. A collimator at the end of the beam line sets the beam spot to 1x2 $mm^2$; the charge flown during the measurement is measured by integrating the beam current on a graphite Faraday cup positioned just behind the samples. The movement of the samples on the x–y axes (perpendicular to beam direction) and the change of the samples (reconstruction of the time sequence) by rotation of the sample holder (i.e. the STRAS polycarbonate filter in this case) are both remotely controlled by the acquisition system. The use of high beam currents (from tens to hundreds of nA) and an optimized X-ray detection system (which includes three Silicon Drift Detectors, one for elements Na-Ca, the other two for elements Ti-Pb), allow for the detection of all the elements with Z>10 with good sensitivity in short measuring times (Lucarelli et al., 2014; Calzolai et al., 2015; Lucarelli et al., 2018; Chiari et al., 2020).

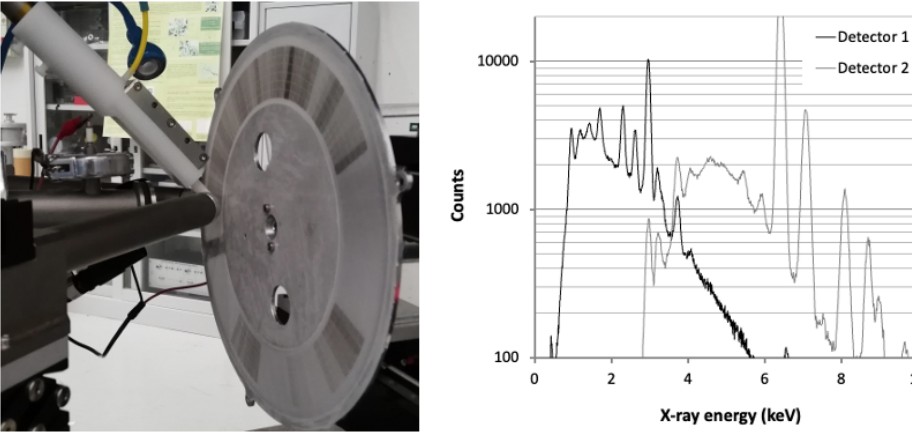

**Figure 2: The STRAS sampler during PIXE analysis at the INFN LABEC external beam setup (on the left). PIXE spectra of a 1-h**
**PM$_{10}$ STRAS sample, simultaneously obtained using a 3 MeV proton beam in 100 s at 150 nA (on the right).**

Several STRAS samples, both blank and loaded filters, were analyzed to evaluate the quality of the spectra and determine the
detection limits (DLs). Elemental concentrations are obtained using the GUPIXWin software (Campbell et al., 2010), using
an H factor free parameter (obtained by the analysis of a set of thin mono/bi-elemental standards by Micromatter Inc). The
total uncertainties on elemental concentrations are determined by the sum of independent uncertainties on certified thicknesses
of the standards (5%), deposit area (5%), air flow (2-5%), X-ray peak fit and peak area counting statistics (2% for the more
abundant elements, then it increases when the concentrations approach DLs). Measurement quality assurance checks were
routinely performed (by analysing the NIST SRM 2783). Measurements and uncertainty estimates were performed according
to "IBA measurements guidelines" (https://www.actris-ecac.eu/pmc-elements.html).
In typical measuring conditions (15 µC of integrated charge per sample, i.e. 60 s at 250 nA proton beam current or 100 s at
150 nA), areal concentration DLs resulted of the order of 10-20 ng cm$^{-2}$ for the low Z elements (Na-Ca) and 1-10 ng cm$^{-2}$ for
medium-high Z elements (Ti-Pb). To obtain DLs for in-air concentrations (ng m$^{-3}$), these values should be simply multiplied
by a factor corresponding to the deposit area (0.9, 1.2 or 1.5 cm$^2$) and divided by the sampled volume (m$^3$).
**4 STRAS validation**
Tests to check the reliability of the new sampler were carried out performing a field campaign where the elemental
concentrations were obtained in parallel by PIXE on STRAS samples (1-h time resolution) and on samples collected by the
Gemini dual channel sampler (Dadolab s.r.l., Italy) operated at 2-h time resolution. PM$_{10}$, PM$_{2.5}$, and PM$_1$ samples were
collected during wintertime 2023 in the University campus of Milan, inside the Dept. of Physics courtyard (45,47665 9,23082;
122 m a.s.l), an urban background site located in the eastern side of the city . To obtain samples with sufficient deposit areal
00 density, the Gemini sampler routinely working with 47 mm diameter membranes was equipped with specially designed inlet

reducers and filter holders, which allow the use of 25 mm diameter filters, thus increasing the areal density by a factor 3.6 (Caiazzo et al., 2021).

To avoid the comparison suffering from effects due to a different type of collecting substrate, one channel of the Gemini sampler was equipped with the same kind of membranes used for STRAS, i.e. 0.8 μm pore size polycarbonate filters. At the same time, to investigate possible PM losses due to a reduced collection efficiency of these membranes, the second channel was equipped with ring-supported thin Teflon filters, which are known to have approx. 100% aerosol retention (Zikovà et al., 2015) and are suitable for PIXE analysis.

For each PM fraction, 20 polycarbonate samples and 20 Teflon samples were collected with 2-h resolution, and 40 STRAS 1h samples were gathered in parallel. All the samples were analyzed at the LABEC external beam PIXE setup as described in the previous section. STRAS hourly elemental concentrations were averaged over a 2-h time interval to be compared with the parallel samples.

## 4.1 *Filter collection efficiency*

It is well-known that the pressure drop on polycarbonate filters depends also on pore size, and bigger pore diameters are generally associated to smaller pressure drops; this can be advantageous especially when a high sampling flow rate throw a small collecting area is considered, but at the same time larger pores are also related to smaller collection efficiencies. As STRAS flow rate and small collecting surface claim for polycarbonate filters with 0.8 μm pores, the efficiency of filters was investigated.

According to the classical theory dating back 1960-1970s (see e.g., Manton, 1978 and 1979; Spurny et al., 1969 and 1972; Chen et al., 2013), the polycarbonate filters can be parametrized through a physical model for which filters are constituted by parallel capillaries; in this way, filter parameters (pore size, filter thickness, and porosity, i.e. ratio of open space over the total filter volume), particle parameters (size and density), and filtration conditions (e.g. the air face velocity on the filter) can be used for the calculation of particle collection efficiency. Particle collection by these filters can be described as the combination of different processes: (1) inertial impact on the filter surface, (2) interception at the pore opening, (3) Brownian diffusion to the pore walls, and (4) Brownian diffusion to the filter surface. The combination of these effects leads to a typical minimum in efficiency (Hinds, 1999) which – in the experimental conditions used in this work – has been calculated to be around 40-50% for particles in the range 50-80 nm assuming a hypothetical density equal to 2 g cm$^{-3}$.

In this work, a field test was performed using parallel samples collected on polycarbonate and Teflon filters for different size fractions. The results are reported in Fig. 3 for PM$_{10}$ which showed the highest aerosol mass concentrations and thus the most reliable values for comparison.

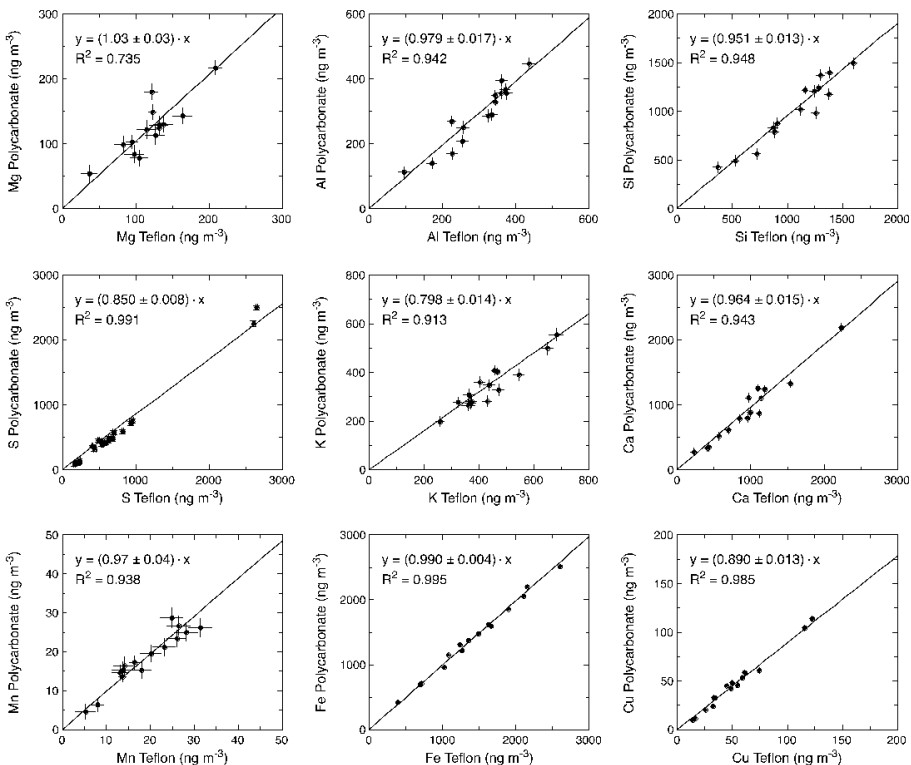

**Figure 3: Scatterplots of elemental concentrations of main elements detected by PIXE in PM$_{10}$, simultaneously collected by the dual channel Gemini sampler on Polycarbonate 0.8 and Teflon membranes. In each plot the fitted correlation line, together with the correlation coefficient, is also shown.**

In general, high correlation was found (> 0.9 for all the elements except Mg) and the comparison between the two datasets shows agreement within few percents for all the detected elements except for S and K, with deviations ranging from 15% to 20% - with samples on polycarbonate 0.8 showing lower values than those on Teflon filters. As aforementioned, for these polycarbonate membranes, minimum in collection efficiency is expected below 100 nm (Hinds, 1999). Thus, only elements with a size distribution strongly skewed towards the Aitken or condensation modes can be affected by significant losses. Focusing on the elements detected with PIXE (Z>10), significant contribution to these modes (at least in urban areas) is expected for S and K only, as S is mainly present as secondary aerosol and K may be produced by biomass burning (see e.g. Bernardoni et al., 2017)

Similar results were obtained for PM2.5 and PM1: focusing on S and K underestimation, slightly lower slopes were found for S (PM2.5: 0.78 ± 0.02, PM1: 0.74 ± 0.05) while the same values within uncertainties were found for K (PM2.5: 0.84 ± 0.03, PM1: 0.80 ± 0.05). As for PM10, no effect was observed for the other elements.

Therefore, it should be taken into account that the polycarbonate membrane with 0.8 μm pores can likely have a lower particle retention for some elements; the lower efficiency of polycarbonate membranes with respect to Teflon filters is also confirmed by other literature studies (Soo et al., 2016; and therein cited literature).

 **4.2 *STRAS field validation***

The STRAS validation on the field was performed using samples collected in parallel on the same filter type to avoid as much
as possible differences in the samples used for comparison.
In Fig. 4, STRAS versus Gemini polycarbonate scatterplots are reported, for main detected elements in $PM_{10}$ samples. As can
be seen from the fitting results, correlations are very good ($\geq 0.9$) for all the elements and the angular coefficients are close to
one (equal to 1 within 10% for all the elements, better than 5% for most of them). For some elements, Gemini data are affected
by larger uncertainties due to the lower aerosol areal concentration on the samples collected with it (this holds true also for
Figure 5 and 6).

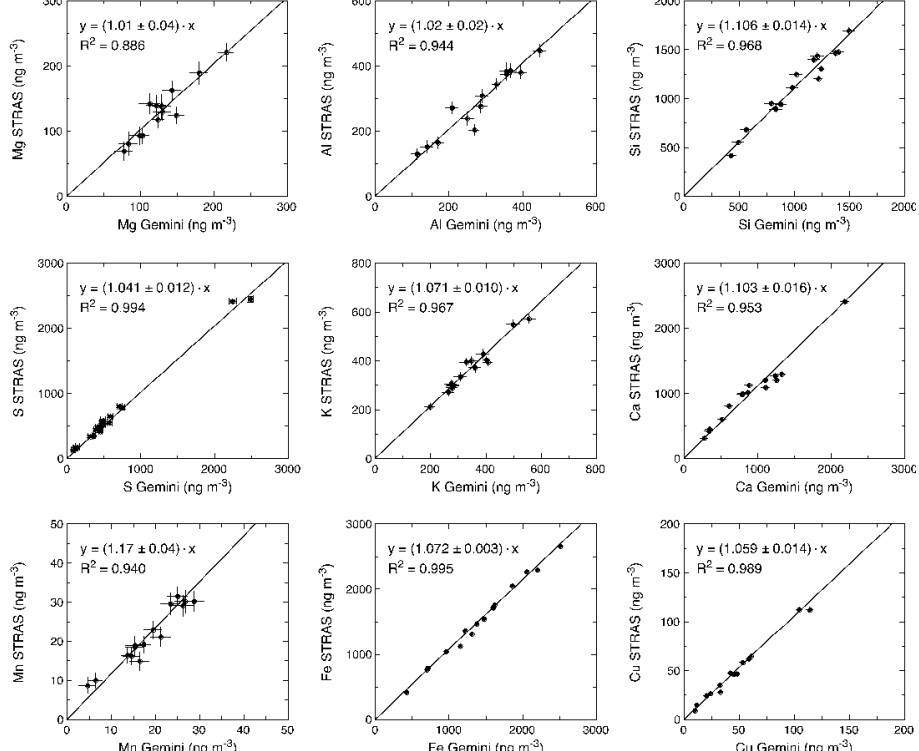


**Figure 4: Scatterplots of elemental concentrations of main elements detected by PIXE in $PM_{10}$ samples simultaneously collected by**
**STRAS and Gemini on polycarbonate (pore size: 0.8 μm). In each plot the fitted correlation line, together with the correlation**
**coefficient, is also shown.**

Similar results were obtained for $PM_{2.5}$ and $PM_1$ (Figures 5 and 6), albeit with a more scattered behaviour due to the lower
counting statistic in fluorescence X-ray peaks (note the difference in scale ranges for $PM_{10}$).

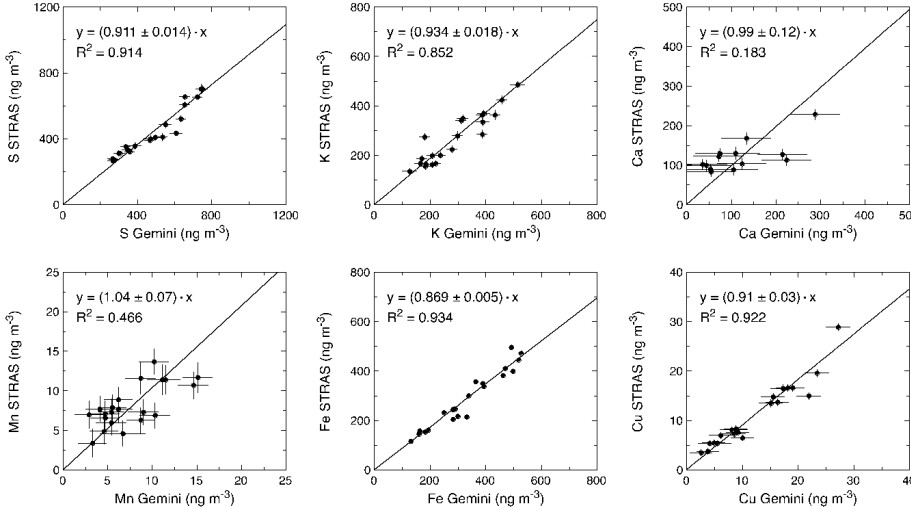

Figure 5: Scatterplots of elemental concentrations of main elements detected by PIXE in PM$_{2.5}$ samples simultaneously collected by STRAS and Gemini on polycarbonate (pore size: 0.8 μm). In each plot the fitted correlation line, together with the correlation coefficient, is also shown.

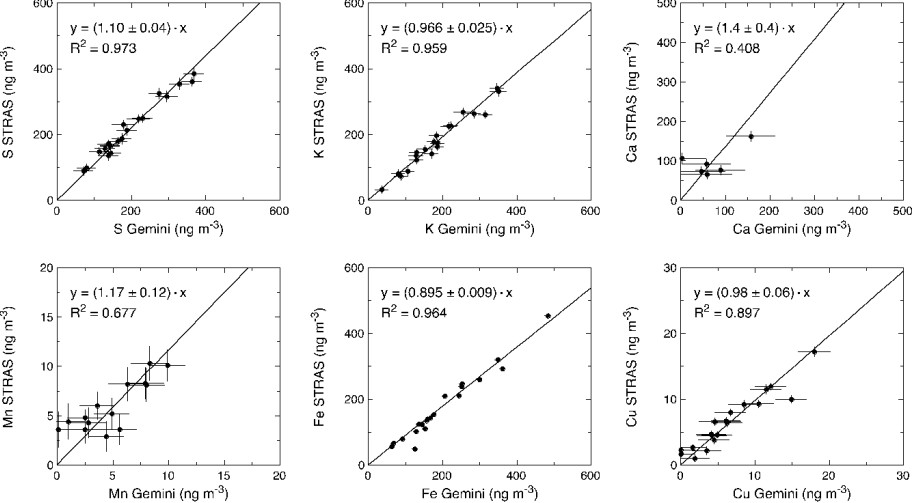

Figure 6: Scatterplots of elemental concentrations of main elements detected by PIXE in PM$_1$ samples simultaneously collected by STRAS and Gemini on polycarbonate (pore size: 0.8 μm). In each plot the fitted correlation line, together with the correlation coefficient, is also shown.

The sampler was used to collect PM$_1$ during the RHAPS (Redox-activity and Health-effects of Atmospheric Primary and Secondary aerosol) project (Costabile et al. 2022; Crova et al., 2024). Within this project, PM$_1$ samples were simultaneously collected by STRAS with 1-h time resolution and by a GEMINI dual channel sampler with daily time resolution, on ring-supported thin Teflon filters (Pall R2PJ047, i.e. the same as those used in the afore described tests but with standard 47 mm

diameter). This allowed a further comparison between the two samplers, although only on daily averages and with samples
collected on different membrane types. Results (Fig. 7) confirmed the outcomes of the tests reported above: elemental
concentrations obtained by the analysis of STRAS samples agree with those obtained by the analysis of 24-h PM samples
collected on ring-supported Teflon filters, except for S and K, that in this case resulted underestimated by about 30%.

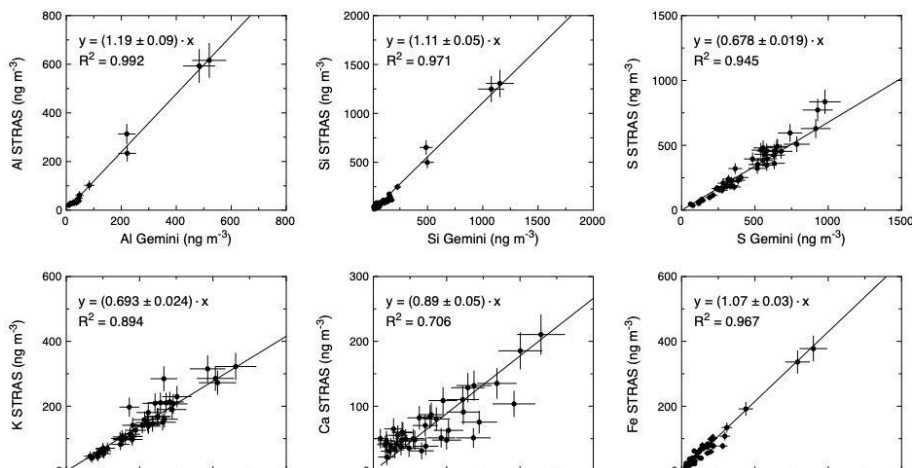


**Figure 7: Scatterplots of elemental concentrations of main elements detected by PIXE in PM$_1$ samples collected in parallel by STRAS**
**(on polycarbonate 0.8 μm membranes) and by Gemini (on ring supported Teflon filters). STRAS concentrations are daily averaged**
**to match the Gemini time resolution. In each plot the fitted correlation line, together with the correlation coefficient, is also shown.**
**5 Conclusions**
A new high time resolution sampler, STRAS (*Size and Time Resolved Aerosol Sampler*), has been designed, developed and
tested.
It has been conceived to collect optimal samples for subsequent PIXE analysis: the aerosol deposit is concentrated on a small
surface area, thus increasing the sensitivity of the technique, on a polycarbonate membrane, which is one of the best choices
for PIXE. Compared to the streaker sampler, formerly used for retrieving 1-h resolution elemental concentrations by PIXE,
the PM areal density is increased by a factor ranging from 1.8 to 3.
STRAS has the advantage of working at the standard flow rate of 1 m$^3$ h$^{-1}$ (US standard flowrate for PM$_{10}$ and PM$_{2.5}$ inlets) so
that it can be equipped with commercial inlets for the collection of main PM fractions (PM$_{10}$, PM$_{2.5}$ and PM$_1$). It allows
automatic sequential sampling of up to 168 hourly samples (1 week), and it is mechanically robust, compact, and transportable.
Samples collected by STRAS can be easily analyzed by PIXE. No sample pre-treatment is needed. Using an optimized set-up
00  (high beam currents and balanced X-ray detector system) one minute for each spot is enough to obtain good statistics: hourly
01  samples collected in 1 week can be analyzed in 3 hours of beam time.

An extensive series of tests was done to verify the correct functioning of STRAS, through comparison with a sequential sampler used for routine PM collection (Gemini dual channel sampler). If the same membrane (0.8 μm pore size polycarbonate) is used by both samplers, the concentrations of all detected elements agree within 10%, thus demonstrating the reliability of the STRAS sampler.

Possible PM losses due to a reduced collection efficiency of these membranes were investigated by simultaneous sampling on ring-supported thin Teflon filters: significant effects (15-30% underestimation) were observed only for S and K, which are elements typically related to smaller particles originated by secondary aerosol processes and biomass burning, respectively (and thus with size distributions which extend down to below 100 nm, where the polycarbonate 0.8 membranes have a minimum in collection efficiency). As a consequence, it is important to be aware that S and K concentrations can be underestimated when using these membranes. In our tests these underestimations were found to be quite stable; however, some parallel sampling, even on a daily basis, may be helpful in determining correction coefficients for other sampling sites. It should be noted that the values found in this work can be considered maximum limits of underestimation since the tests were carried on in places characterized by a strong prevalence of ultrafine aerosols, such as secondary and combustion ones.

Information obtained by STRAS plus PIXE, i.e. the elemental composition (from Na to Pb), may be complemented by other measurements. The same STRAS samples can be analyzed by other IBA techniques (simultaneously with PIXE) and by optical techniques (before PIXE analysis, to avoid possible changes of the optical properties of the samples due to sample-beam interaction). In particular, Black Carbon and Brown Carbon concentrations can be measured by multi-wavelength optical analysis (Bernardoni et al., 2017; Massabò et al., 2015).

Furthermore, STRAS may be used "side-by-side" with other high-time resolution instrumentation, like e.g., the Aerosol Chemical Speciation Monitor (ACSM) for the non-refractory components ($NO_3^-$, $SO_4^{2-}$, $NH_4^+$, $Cl^-$, OA), the Aethalometer for black carbon, and the on–line Sunset field analyzer for organic and elemental carbon (OC and EC).

A two-stage version of STRAS is under development with the fine fraction collected on a polycarbonate filter, as in the model described in this paper, while the coarse fraction will be collected by inertial impaction on a thin polypropylene foil.

**Author contribution**

Conceptualization of the new sampler: SN, RV, PP, VB, GC, DM, GV and FL. Mechanical and electronical implementation of the sampler: LCar, CC, MM. Aerosol samplings: VB, LCad, FC, AF, FG, GV. PIXE measurements: SN, GC, MC, FG, FL Formal analysis: SN, FG, MC. Data analysis, validation and discussion: SN, RV, PP, VB, GC, MC, DM, FM, LP, GV, VV, FL. Project Management and fund raising: FL, GC, RV. Original draft writing: SN, RV, GC. Paper review and editing: RV, PP, VB, MC, FC, CF, DM, FM, GV and FL.

**Competing interests**
The contact author has declared that none of the authors has any competing interests.
**Acknowledgements**
The authors kindly acknowledge the financial support of INFN, through the experiment TRACCIA and the grant
STRASPEED, and the MIUR project RHAPS (grant number 2017MSN7M8). This work was realized with the contribution of
Fondazione CR Firenze. The work was supported by IR0000032 – ITINERIS, Italian Integrated Environmental Research
Infrastructures System (D.D. n. 130/2022 - CUP B53C22002150006) Funded by EU - Next Generation EU PNRR- Mission 4
"Education and Research" - Component 2: "From research to business" - Investment 3.1: "Fund for the realisation of an
integrated system of research and innovation infrastructures". The work was supported also by the Italian PON "Ricerca e
Innovazione 2014-2020" project PER-ACRIS-IT (PIR_01_00015). Finally, the authors acknowledge the fruitful collaboration
with DadoLab srl, and in particular with G. Cazzuli, G. Gargioni and S. Alberti.

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
