# Peer review of "STRAS: a new high time resolution aerosol sampler for PIXE analysis"

_Atmospheric Measurement Techniques, 2024_

## Author Comment (AC1)

**Answer to RC1**

The manuscript titled "STRAS: a new high time resolution aerosol sampler for PIXE analysis" has developed a new hourly sampler, called STRAS, which can be used to automatically collect sequential samples of up to 168. Specially, a small surface area for depositing particles is designed for detecting elements using the Particle Induced X-ray Emission (PIXE) technique. Field measurements were conducted to perform comparison between STRAS and other sampling instruments. Overall, it is a good advancement of the sampling technique. There are several comments that need to be addressed as below.

The main issue is how to verify the accuracy and reproductivity of the newly invented sampler? The manuscript has compared the results between STRAS and Gemini, which both shared the same inlet, cabinet, pump, air flow control system as stated by the authors. As samplers are subject to various uncertainties and Gemini is not a reference sampler, the good consistency between STRAS and Gemini doesn't guarantee the reliability of STRAS. It is essential to use standard particles with known concentrations and compositions for verifying a new instrument.

Gemini and STRAS share the same inlet, cabinet, pump and air flow control system, but the sampling chamber and filter holder are extremely different, and these are the critical points we wanted to test. With a high-resolution continuous sampler, like STRAS, the critical point is the forcing of the air flow through a small surface on a filter, which has to move every hour.

In general, for a single mode sampler, the important aspects are the inlet and the flowrate, which together determine the cut-off diameter, and the absence of leakages (both lack of sealing and loss of particles on the internal surfaces).

In both Gemini and STRAS samplers the same commercial inlets are mounted in order to guarantee equivalent cut-off performances for PM10, PM2.5 and PM1 when operating at a flow of 1.0 m3/h; the flow is controlled and measured, and remains stable within 2-3%. These inlets are commercial ones; for PM10 and PM2.5 they are made according to CEN design (EN 12341:1998 and EN 14907:2005) but remodulated to work at a flow of 1m3/h, while for PM1 there is no international standard and therefore they have been designed - as do all those who sell PM1 sampling inlets - using the inertial impact theory.

As for Gemini, it may be considered as a reference sampler as it complies with EN 12341:2023 and it has just successfully completed all the tests conducted by TÜV Rheinland. As for STRAS, the aspects related to leakages were instead to be tested and for this reason the measurements described in this article are relevant. Possible air flow losses and loss of particles on surfaces were minimized in the design of the sampler (e.g., sampling lines are realized in anodized aluminum, a widely used conductive material which reduces wall losses which might affect especially small sized particles) and tests were performed to prove it by comparison with Gemini as described in the paper. We agree that the use of standard particles is important especially for multistage samplers, and we already plan to use them to test the advanced two-stages version of STRAS.

Section 4.1, Since the authors have revealed the filter collection efficiency, is this collection efficiency robust or random? how to account for the loss of particles in the application?

This is an important point.

The 0.8 micron membranes have a minimum in collection efficiency below 100 nm (Hinds, 1999). Thus, filter collection efficiency is not random, but it depends on the size distribution of the particles: only elements with a size distribution strongly skewed towards the Aitken or condensation modes can be affected by significant losses. Focusing on the elements detected with PIXE (Z>10), significant contribution to the Aitken and accumulation mode (at least in urban areas) is expected for S and K only, as S is mainly present as secondary aerosol and K may be produced by biomass burning (see e.g. Bernardoni et al., 2017). In the measurements presented in this work and from previous measurements, a fair reproducibility is observed: the loss is always between 15 and 30% for S and between 20 and 30% for K. Conversely, for crustal and marine elements the effect is expected and proved to be minimal, as a result of the measurements we made to evaluate these effects. This allows a reasonable evaluation of the effect due to the collection efficiency and the correction of the results. We also want to point out that this is a membrane type issue. STRAS can also work with the 0.4 micron pore size polycarbonate membranes, and with these the collection efficiency is close to 100%.

We have added in the conclusions of the text the following recommendations on how to account for these effects:

"Possible PM losses due to a reduced collection efficiency of these membranes were investigated by simultaneous sampling on ring-supported thin Teflon filters: significant effects (15-30% underestimation) were observed only for S and K, which are elements typically related to smaller particles originated by secondary aerosol processes and biomass burning, respectively (and thus with size distributions which extend down to below 100 nm, where the polycarbonate 0.8 membranes have a minimum in collection efficiency). As a consequence, it is important to be aware that S and K concentrations can be underestimated when using these membranes. In our tests these underestimations were found to be quite stable; however, some parallel sampling, even on a daily basis, may be helpful in determining correction coefficients for other sampling sites. It should be noted that the values found in this work can be considered maximum limits of underestimation since the tests were carried on in places characterized by a strong prevalence of ultrafine aerosols, such as secondary and combustion ones."

Line 95 - 105: The description is not written in a scientific manner, it is more like a manual.

This part has been rewritten as follows:

The sampler has been conceived and designed in order to obtain a high-time resolution (modulable from 30 min to a few hours) PM10, PM2.5, and PM1 collection, on "small" (about 1 cm2) deposit areas, with automatic sequential sampling of at least 168 samples (1 week of hourly samples), to be mechanically robust, easy to use, compact and easily transportable

Line 288 – 289: I cannot foresee the application of STRAS in measurement of black carbon or brown carbon as aethalometer has advantages especially on its time resolution.

Indeed STRAS may be used "side-by-side" with the aethalometer. However, the possibility of off-line black carbon and brown carbon measurements on the filters collected with STRAS has its advantages. First of all, there is no need for a different instrument (i.e. an Aethalometer) running in parallel to STRAS sampling. Further, laboratory instrumentation (see e.g. the polar photometer PP_UNIMI (Bernardoni et al., 2017) and the multi-wavelength absorbance analyzer MWAA (Massabò et al., 2015)) measuring both transmitted and scattered light and applying the same radiative transfer scheme as the one used by the Multi-Angle Absorption Analyzer (MAAP; Petzold and Schonlinner, 2004) can provide the information on the aerosol absorption coefficient at different wavelengths (and consequently on BC and BrC) with no need of loading corrections and assumptions on multiple scattering enhancement parameters.

---

## Author Comment (AC2)

**Answer to RC2**
General comments:

This paper introduces a new aerosol sampler for observing particulate matter composition capable of taking many samples with high temporal resolution, offering the potential for more efficient atmospheric monitoring—an increasingly critical need. However, the manuscript lacks several key elements, including a quantitative evaluation and thorough discussion of the sampling method, as well as a comparison of its advantages and disadvantages relative to conventional methods. These points require further elaboration.

We thank the referee for raising these issues, discussed in more detail in the following. We have accordingly changed the paper.

A major concern is the observed loss of fine particles for certain elements compared to Teflon filters. This raises questions about the suitability of the method for fine particle sampling ($PM_{2.5}$, $PM_1$). If the sampler is intended for fine particle applications, the authors should clearly discuss its validity and propose solutions to address these issues.

For nuclepore with 0.8 micron pores, minimum in collection efficiency is expected below 100 nm (Hinds, 1999). Thus, even sampling fine particles, only elements with a size distribution strongly skewed towards the Aitken or condensation modes can be affected by significant losses. Focusing on the elements detected with PIXE (Z>10), significant contribution to the Aitken and accumulation mode (at least in urban areas) is expected for S and K only, as S is mainly present as secondary aerosol and K may be produced by biomass burning (see e.g. Bernardoni et al., 2017). In the measurements presented in this work and from previous measurements, a fair reproducibility is observed: the loss is always between 15 and 30% for S and between 20 and 30% for K. Conversely, the effect is expected and proved to be minimal for crustal and marine elements, as a result of the measurements we made to evaluate these effects. This allows a reasonable evaluation of these effects due to the collection efficiency and the correction of the results.
We also want to point out that this is a membrane type issue. STRAS can also work with the 0.4 micron pore size polycarbonate membranes, and with these the collection efficiency is close to 100%.

We have added in the conclusions of the text the following recommendations on how to account for these effects:

"Possible PM losses due to a reduced collection efficiency of these membranes were investigated by simultaneous sampling on ring-supported thin Teflon filters: significant effects (15-30% underestimation) were observed only for S and K, which are elements typically related to smaller particles originated by secondary aerosol processes and biomass burning, respectively (and thus with size distributions which extend down to below 100 nm, where the polycarbonate 0.8 membranes have a minimum in collection efficiency). As a consequence, it is important to be aware that S and K concentrations can be underestimated when using these membranes. In our tests these underestimations were found to be quite stable; however, some parallel sampling, even on a daily basis, may be helpful in determining correction coefficients for other sampling sites. It should be noted that the values found in this work can be considered maximum limits of underestimation since the tests were carried on in places characterized by a strong prevalence of ultrafine aerosols, such as secondary and combustion ones."

Additionally, it was difficult to understand the superiority of STRAS over the conventional streaker sampler and the Gemini sampler used for this comparison. I expect that STRAS allows the collection of a larger number of samples and a larger amounts of particles in one spot, but there is insufficient data regarding the Germini sampler's capacity, making it difficult for readers to draw meaningful comparisons. I would recommend to make a table to explain merits and limitations of each sampling method together with other methods mentioned in the introduction.

We have explained more clearly in the paper the differences and advantages of STRAS with respect to the STREAKER and the conventional sequential samplers like Gemini (lines 97-102).

Advantages of STRAS against STREAKER:
- higher areal load, and so higher sensitivity for PIXE
- flexibility in time-resolution
- possibility to work directly with commercial US-EPA inlets
- STREAKER sampler is not produced anymore

-data logging for sampling parameters (air flow, pressure drop, elapsed time etc..) and for sample position (the position of every single spot is registered and can be reproduced also in case of black-outs or stepper motor malfunctioning). STREAKER registered the sampling parameters (in a far less user-friendly mode with respect to STRAS) but not the sample positioning data. [This information was missing: we added it at lines 165-167 in the sampler description].

- better separation of the sample deposits (i.e. sampling slots) avoiding the moving average due to the continuous frame rotation present in the STREAKER sampler

Advantages of STRAS against the GEMINI:
- higher areal load, and so higher sensitivity for PIXE
- flexibility in time-resolution (down to 1h)
- the number of samples that can be collected with no need of filter change or operator intervention: GEMINI can perform up to 21 samples without any operator intervention (less than a day on hourly basis); STRAS can collect up to 168 samples without any need of changing filters, allowing 1 full week of samplings on a hourly basis with the instrument unattended. While unattended, both GEMINI and STRAS can be controlled remotely.

Specific comments

1.  Introduction:

·  As noted in the general comments, please consider adding a table to summarize the merits and

limitations of different sampling and analytical methods. In addition, include additional details, such as the differences in detection limits, applicable elements, costs, etc., between EDXRF and PIXE, and the advantages of STRAS sampling system over traditional streaker samplers. This would enhance the clarity of the introduction.

We agree with the referee that a direct comparison of the pros and cons of the different sampling techniques may help and guide the reader. Therefore, we added more information in the paper, at the end of the introduction section, in a discursive way, which we felt was clearer than a table (lines 97-102). The paper focuses on a new sampling device, therefore a thorough discussion of the analytical techniques for the determination of aerosol composition is beyond the scope of this work. Nevertheless, we added an explanation for having designed a sampler "dedicated" to PIXE analysis and added some references for the comparison of the different analytical techniques (lines 103-109).

1.  Sampler design:

·  Please include information about the size and weight of STRAS sampler, since you mentioned that

the sampler was designed to be "compact and easily transportable."

We have added this information to the paper, lines 171-172.

·  Please provide details on how particle size resolution is achieved. There was no information on size

separation. In addition, it was not clear for me whether the sampler can collect different size fractions at a same time or not.

The STRAS version described in this paper is a single stage sampler. It collects PM by filtration on polycarbonate membranes. It can be equipped with commercial single cut-off inlets designed to select

PM10, PM2.5 or PM1 at 1 m3/h. So, depending on the mounted inlet it can collect one of these fractions (one at a time).
An implementation of STRAS is currently in progress to simultaneously collect both the fine and coarse PM fractions as previously done by the streaker samplers, but it is not described in this paper.

1. PIXE analysis of STRAS samples

・ P6. L174: How did you determine the errors (especially for deposit area)? If they were based on analytical data or references, include this information.

We added the following information (lines 201-203):

Measurement quality assurance checks were routinely performed (by analysing the NIST SRM 2783). Measurements and uncertainty estimates were performed according to "IBA measurements guidelines" (https://www.actris-ecac.eu/pmc-elements.html).

As for the deposit area, its size is determined by the dimensions of the sucking orifice, which is obtained by 3D print. The used printer has an accuracy of less than 1 tenth of millimetre. A slightly more conservative estimate of the uncertainty was taken into account due to a possible broadening of the deposit on the sides (not evidenced by the careful measurements performed on the deposit spots).
https://www.actris-ecac.eu/pmc-elements.html

・ P6. L177: If filters were washed with acid, would this reduce the blank values? Trace metal concentrations can be as low as several ng/m$^3$ in remote areas like marine atmosphere.

For PIXE analysis, with clean filters as Polycarbonate, or Kapton/Teflon/Polypropylene…, the main limitation to detection limits (DLs) comes from the thickness of the filters as this increases the continuum background in the spectra due to secondary bremsstrahlung and Compton scattering. For this reason, in the past, we have deeply characterized the DLs with the different collecting substrata, as reported in Calzolai et al., 2015.

Polycarbonate membranes are very clean: rarely we have observed small contaminations of Si and Br in Polycarbonate filters, but these batches have been discarded.

1. STRAS validation

・ P7. L184: Please specify the sampling position (longitude, latitude) and height.

The sampling campaign was performed inside the Dept. of Physics courtyard, an urban background site located in the eastern side of Milan (45,47665 9,23082; 122 m a.s.l). We have added this information to the paper.

・ 4.1: You mentioned potential underestimation of S and K due to their enrichment in fine particles. However, you don't address any solutions or recommendations for it. In addition, I don't understand why other elements have good agreement even in PM$_1$ (Fig. 7) despite that they are all fine particles.

As aforementioned, for nuclepore with 0.8um pores, minimum in collection efficiency is expected below 100 nm. Thus, even sampling fine particles, only elements with a size distribution strongly skewed towards the Aitken or condensation modes can be affected by significant losses. Therefore, most of the elements have good agreement whatever the sampled PM fraction (PM10 or PM2.5 or PM1). We added some recommendations in the paper on how to handle this effect.

・ 4.2: Please explain why Germini data has larger errors (for example, Figure 5, 6). Is this due to the smaller sample quantities on the filter spots?

Yes, it is due to the lower areal densities on Gemini samples. We added this information in the paper (lines 263-265).

・ Figure 7: Why Ca concentrations at low levels are higher in STRAS sampler compared to the Germini sampler?

It may be due to an unperfect spectra fitting at low concentrations, due to the subtraction of the continuum background due to Bremsstrahlung in the spectra. Nevertheless, the effect is small (please note the very high uncertainties and the y axis range)

1. Conclusion

・ P12 L284: This should be mentioned in section 4. Also please refer some papers here.

Yes, we added this information in section 4 (with references).

Technical comments

・ P2. L46: optimal solution ⇒ optimal analytical solution

Yes, we did it.

・ P3. L93-94: I don't think the sentence "Furthermore…samplers." is necessary in the introduction, this should be moved to conclusion or discussion.

Yes, we moved it in the conclusion section.

・ P8. L224: How about adding "(e.g. S and K)" at the end of the sentence?

Yes, we did it.

---

## Author Comment (AC3)

**Answer to RC3**

This manuscript describes the STRASS equipment, an aerosol sampler for the collection of atmospheric particles. Sampling is performed in short periods of one hour, over a polycarbonate filter surface; that permits the collection of 168 samples, sequentially and automatically, during one week. The exposed filter is sequentially transported to the laboratory, where the direct measurement of elementary aerosol composition by PIXE is performed, without further treatment. This sampling/PIXE analysis methodology permits the evaluation of trace aerosol composition with hourly discrimination at acceptable costs and is therefore potentially interesting for source apportionment purposes.

The STRASS sampler was built to replace and improve the previous STREAKER sampler used for the same objectives, which is not commercially available, anymore.

In the paper the characteristics and performance of the STRASS sampler are evaluated with various experiments to validate the sampler and polycarbonate 0.8 μm pore membrane capability to collect correctly PM1, PM2.5 and PM10 aerosols.

The evaluation of 0.8 μm pore polycarbonate filter was done by parallel sampling with a Teflon membrane filter recognized as having high filter efficiency for submicrometric particles. The results, exposed in Figure 3, show very similar concentration values. However, the presented results are for PM10 and elements usually associated with coarse particles (with possible exception of S). Therefore, from the experiment, it is not clearly demonstrated the capability of the 0.8 μm pore membrane to collect efficiently fine particles. Results for elements principally associated with fine particles and for PM1 or PM2.5 fractions would be more relevant to the evaluation of filter collection efficiency.

For nuclepore with 0.8 micron pores, minimum in collection efficiency is expected below 100 nm (Hinds, 1999). Thus, even sampling fine particles, only elements with a size distribution strongly skewed towards the Aitken or condensation modes can be affected by significant losses.
Focusing on the elements detected with PIXE (Z>10), significant contribution to the Aitken and accumulation mode (at least in urban areas) is expected for S and K only, as S is mainly present as secondary aerosol and K may be produced by biomass burning (see e.g. Bernardoni et al., 2017). In the measurements presented in this work and from previous measurements, a fair reproducibility is observed: the loss is always between 15 and 30% for S and between 20 and 30% for K. Conversely, the effect is expected and proved to be minimal for crustal and marine elements, as a result of the measurements we made to evaluate these effects.
We did not report PM2.5 and PM1 scatterplots as we obtained similar results with PM10, however we agree that this information may be of interest and we added it in the paper (a slight effect can be observed for S).
We also want to point out that this is a membrane type issue. STRAS can also work with the 0.4 micron pore size polycarbonate membranes, and with these the collection efficiency is close to 100%.

The reliability and accuracy of the STRASS sampler was done in a field validation, by parallel sampling in the external environment with the STRASS and a GEMINI sampler, using both the same polycarbonate 0.8 μm pore membranes. The results show remarkable similarities for both samplers, principally for PM10, a clear indication of the reliability of the STRASS sampler which seems not to suffer from flow control or leak problems.

The collection capability of 0.8 μm pore membrane filters is further evaluated in field conditions by using parallel sampling with the GEMINI sampler provided with Teflon membrane filters. The results presented in Figure7 for PM1 particles show evidences of inefficient polycarbonate membrane filtration for elements concentrated in submicrometric size ranges, such as S and K.

The results of this campaign confirm the results obtained in the courtyard tests: a loss for S and K, no loss for the other elements. The losses for S and K are slightly different from those found in the tests (about 30% for both elements versus 15% for S and 20% for K). Nevertheless, it should be noted that these data were obtained in a real field campaign, where the sampling difference is not only the type of membrane: in particular, in this case, STRAS was used to sample on an hourly basis while GEMINI

was conventionally used on a daily basis. Further, the campaign was performed in a different sampling site, Bologna, where the S and K size distributions may be different from Milan.

We have added in the conclusions of the text the following recommendations on how to account for these effects:

"Possible PM losses due to a reduced collection efficiency of these membranes were investigated by simultaneous sampling on ring-supported thin Teflon filters: significant effects (15-30% underestimation) were observed only for S and K, which are elements typically related to smaller particles originated by secondary aerosol processes and biomass burning, respectively (and thus with size distributions which extend down to below 100 nm, where the polycarbonate 0.8 membranes have a minimum in collection efficiency). As a consequence, it is important to be aware that S and K concentrations can be underestimated when using these membranes. In our tests these underestimations were found to be quite stable; however, some parallel sampling, even on a daily basis, may be helpful in determining correction coefficients for other sampling sites. It should be noted that the values found in this work can be considered maximum limits of underestimation since the tests were carried on in places characterized by a strong prevalence of ultrafine aerosols, such as secondary and combustion ones."

Contrary to previous figures, figure 7 shows scatterplots with X and Y in logarithmic scales. From the figure it seems that presented linear regressions and correlation coefficients are also for the logarithm of concentrations. This presentation permits the evidencing of lower concentration values, that, anyway, are already strongly influenced by PIXE detection limits and filter blanks variability. However, visualization of differences at higher concentration ranges are reduced. A parallel figure using linear scales would be useful for a more clear evaluation of polycarbonate filters performance.

Scatterplots of figure 7 are presented in log scales but the linear regression parameters and correlation coefficients are calculated on the original concentrations (not log). We chose log plots due to the larger range of variability of these data; nevertheless, we agree with the referee about the limitations of this visualization. In order to uniform the paper plots, we have decided to change figure 7 to linear plots, as the ones reported in the following: